# Multiparameter Monitoring with a Wearable Cardioverter Defibrillator

**DOI:** 10.3390/s22010022

**Published:** 2021-12-21

**Authors:** Ursula Rohrer, Martin Manninger, Andreas Zirlik, Daniel Scherr

**Affiliations:** Division of Cardiology, Department of Medicine, Medical University of Graz, 8036 Graz, Austria; u.rohrer@medunigraz.at (U.R.); martin.manninger-wuenscher@medunigraz.at (M.M.); andreas.zirlik@medunigraz.at (A.Z.)

**Keywords:** defibrillator, sudden cardiac death, ventricular arrhythmia, telemedicine, monitoring, heart failure

## Abstract

A wearable cardioverter-defibrillator (WCD) is a temporary treatment option for patients at high risk for sudden cardiac death (SCD) and for patients who are temporarily not candidates for an implantable cardioverter defibrillator (ICD). In addition, the need for telemedical concepts in the detection and treatment of heart failure (HF) and its arrhythmias is growing. The WCD has evolved from a shock device detecting malignant ventricular arrhythmias (VA) and treating them with shocks to a heart-failure-monitoring device that captures physical activity and cardioacoustic biomarkers as surrogate parameters for HF to help the treating physician surveil and guide the HF therapy of each individual patient. In addition to its important role in preventing SCD, the WCD could become an important tool in heart failure treatment by helping prevent HF events by detecting imminent decompensation via remote monitoring and monitoring therapy success.

## 1. Introduction

According to data from the World Health Organization, more than 17 million people per year die due to cardiovascular diseases (CVD), making this entity responsible for every third death worldwide. Within these CVD-related deaths, around 25% is categorized as sudden cardiac death (SCD) [1,2].

To prevent SCD, current guidelines recommend the optimization of cardiovascular (CV) risk factors in primary and secondary prevention. The second therapeutic principle is used to assess the individual risk of SCD in patients with cardiomyopathies by taking risk factors and comorbidities into account [2,3,4].

The most effective long-term therapy used to prevent SCD in high-risk patients is an implantable cardioverter defibrillator (ICD) [5,6,7]. The benefit derived from an ICD strongly depends on adequate risk assessment beforehand.

Thus, the implantation of an ICD should be reserved for patients with a permanent high risk for SCD rather than those with a potentially reversible SCD risk. The latter group of patients with a temporary risk for SCD may be candidates for a wearable cardioverter defibrillator (WCD). The WCD is also a viable option for patients waiting for ICD implantation or patients after ICD explantation, i.e., due to infections or endocarditis.

## 2. The Wearable Cardioverter Defibrillator (WCD)

Over twenty years ago, the wearable cardioverter defibrillator (WCD) was developed to find a potential solution to protecting patients with a temporary high SCD risk or with a permanent risk for SCD who are not eligible for immediate ICD implantation [8,9,10].

### 2.1. Composition of the Device

A wearable cardioverter defibrillator consists of a monitor with a rechargeable battery and a fabric vest provided with four ECG electrodes to monitor the patient’s heart rhythm as well as defibrillation electrodes to deliver electrical shock therapy, if needed. Additionally, the WCD captures a variety of information that is transmitted to an online network. These data include basic information such as wearing time, alarms, treatments, the patient’s activity level, body position, and heart failure monitoring via cardioacoustic biomarkers analyzed using a more complex algorithm that is further explained in Section 3.3.

#### 2.1.1. The Fabric Vest

The device consists of several components, starting with a fabric garment that comes in different sizes and has an adjustable belt to fit every body size tightly and a back part with shoulder straps to attach technical components such as the defibrillation pads, the electrode belt, and the vibration box (see Figure 1).

#### 2.1.2. The Defibrillation Pads

The three defibrillation pads are in contact to the patient’s skin: one pad that includes the heart sounds sensor (see Section 3.3—Monitoring Heart Failure) is located in an apical position, and the other two pads are located in a posterior position on both sides of the vertebra and fitted closely to the dorsal muscles.

The pads are dry and not adhesive and include ten gel capsules (see Figure 1). Shortly before delivering a WCD shock, an acoustic and visual alarm via the monitor and a vibrational alarm via the vibration box on the patient’s back inform about the imminent shock. Then, blue contact gel from the gel capsules is automatically applied to enhance the transmission of the electrical current. The energy of the biphasic direct current shock can be individually adjusted from 75 to 150 J.

#### 2.1.3. The ECG Electrodes

The WCD continuously monitors and analyzes the heart rhythm of a two-lead ECG with four ECG electrodes appearing as an electrode belt attached to the fabric harness (see Figure 2). The electrodes capture a surface ECG but does not include information about the atrial and the ventricular activation separately; therefore, the arrhythmia detection algorithm is kept simple [11]: the heart rate (HR) is continuously assessed automatically by applying the Fourier transformation frequency plot on the QRS complexes detected from two ECG channels that are provided from the front-to-back and right-to-left surface ECG leads. The preset heart rate zones are 150 bpm for ventricular tachycardia (VT) and 200 bpm for ventricular fibrillation (VF) but can be adjusted individually by the treating physician.

When the heart rate exceeds one of the pre-set thresholds, a QRS analysis is initiated, including an analysis of stability, onset, and QRS morphology in a second step, to distinguish whether a ventricular arrhythmia (VA) is present. If the HR differs significantly between the two leads, the HR is weighed lower than the analysis of QRS morphology, onset, and stability. The QRS morphology is matched with a template ECG in the patient’s normal rhythm. If only a part of the analysis is applicable due to the quality of the captured ECG, one part of the detection algorithm is weighed higher for the adjudication and vice versa (see Figure 3) [12].

In the case of a conscious and hemodynamically stable patient, the response buttons that sit on the monitor unit can deactivate the imminent shock delivery to avoid inappropriate shocks [12].

The electrode belt also includes a three-axis accelerometer to collect information about the individual daily step count as well as the patient’s body position. While the step count is counted in continuous numbers, the body position is categorized into upright (angle of 60–90 degrees), reclined (angle of 30–60 degrees), or lying (0–30 degrees) positions [13].

#### 2.1.4. The Monitor

The monitor unit comes as a small case with a LCD touch screen, two response buttons—one in the front and one in the back—a connection point for the electrode belt, and a speaker unit for the audible alarms (see Figure 4), and it is usually carried in a small bag provided with the entire device.

The enclosed processor is provided with a software to monitor the ECG and to analyze the captured information to detect and treat ventricular arrhythmias. The lithium-ion batteries have an energy capacity for 24 h continuous operation and up to five shocks, while every patient is provided with two sets of battery packs and a charging station. The LCD monitor shows the current battery status and whether the signal quality of the ECG electrodes is sufficient and has a control panel to send data from the monitor unit to the network manually or to adjust options such as language or volume. The device has three different alarms: physiological alarms that appear as a reaction to arrhythmias; technical alarms when a technical problem is present; and informative alarms when the batteries need to be exchanged soon or the signal quality of the ECG is insufficient because the ECG electrodes are too loose, which is assessed by a microampere alternating current. When any kind of problem is detected, the monitor also provides further instructions and additional information about the triggered alarm [14,15].

### 2.2. Clinical Application

The first data presented about clinical WCD use in patients with either symptomatic heart failure with reduced ejection fraction (WEARIT study) [16] or patients at high risk for SCD after acute myocardial infarction or bypass surgery (BIROAD Study) [16] were gathered in 289 patients and attested to safety and efficacy of the WCD as a treatment option for malignant arrhythmias in these patients. The WEARIT study included 177 patients (LVEF 30 ± 10%, 83% male) with symptomatic heart failure and a LVEF below 30%, with a functional New York Heart Association (NYHA) Class III or IV in an ambulatory setting. The BIROAD study included 112 patients (LVEF 19 ± 7%, 79% male) with recent myocardial infarction meeting one of the following additional inclusion criteria: experiencing a VA within 48 h of the index event or even within 48 h after coronary artery bypass grafting (CABG); a LVEF below 30%; survival from sudden cardiac arrest of syncope at least after 48 h after CABG; and not eligible for ICD implantation, not able to be implanted with an ICD within 4 months due to capacity reasons, or refusal to use an ICD. Compared with the current available WCD, the device used for the WEARIT/BIROAD study used a monophasic waveform with a maximum output of 285 joule and was not provided with a telemonitoring functionality. During follow-up, eight WCD shocks were delivered in 6/289 (2%) patients with a 75% (6/8) shock efficacy. During follow-up, 12/289 (4.2%) were found deceased up while the authors stated that the patients were either not wearing the WCD correctly or not wearing it at all, with only one patient with a detected VA and an ineffective shock due to reversal of the defibrillation pads. Six inappropriate shocks were captured within 901 patient months (0.7% per month of patient use) [16].

Following this clinical non-randomized study, the prospective WEAR-IT II registry was used to evaluate the newer generation of WCD with a lower weight and a bipolar waveform with a maximum energy level of 150 J: the registry was used to investigate 2000 well-compliant patients (median age 62 years, IQR 16; median LVEF 25%, median wear-time 22.5 h/day) at high SCD risk, with 805/2000 (40%) patients suffering from ischemic cardiomyopathy, 927/2000 (46%) with non-ischemic cardiomyopathy (NICMP), and 286/2000 (14%) with a congenital or an inherited heart disease in a real-world setting. After a median WCD wear-time of 90 days, 41/2000 (2%) patients experienced a total of 30 events of VA with shocks and a first shock efficacy of 100%, with 46% of all detected sustained VTs not requiring electrical therapy and a very low inappropriate shock rate of 0.5% (10/2000). After a median follow-up duration of three months, the group of ischemic and congenital cardiomyopathy (CMP) showed the highest rate of VA (3% in ischemic and congenital CMP vs. 1% in non-ischemic CMP, *p* = 0.02). At the end of follow-up, 1160/2000 (58%) patients no longer had an indication for ICD implantation. The WEAR-IT II registry showed the efficacy and safety of the WCD in 2000 high-risk patients in different clinical settings: good patient compliance was already observed when using the newer and therefore lighter model of the WCD; the VA detection was accurate; and the bipolar shocks were effective, with a low rate of inappropriate shocks [17].

Valuable real-world data on WCD have been provided over the last 20 years by nationwide registries, such as the German Registry reporting their experience of 6043 patients in 404 centers [18], the US registry with 3569 patients included [19], and the Austrian WCD registry capturing 448 patients from 56 centers over more than 10 years. Within the Austrian WCD registry, 19 episodes of shocked VT/VF were documented in 11/448 (2.5%) patients, with a first shock success rate of 84% and an all-over shock efficacy of 95%. The rate of inappropriate shocks was 0.4% (2/448 patients) [20].

Several studies focused on different clinical settings such as recent cardiac decompensation in advanced heart failure [21,22] or during the evaluation of a newly diagnosed CMP [23] as well the WCD use in smaller subgroups such as patients with congenital and inherited CMP [24] and patients suffering from peripartum CMP [25,26].

Several groups focused on patients after a recent acute myocardial infarction with or without revascularization [27,28] as these patients not only were a very vulnerable cohort of patients but also showed the highest rate of shocks when compared with other groups [29].

In a German Single Center study of 114 patients (follow-up of 52.0 days (25.0, 90.0), compliance 23.1 h (19.0, 23.8)) consisting of 31.6% of patients with ischemic CMP (ICMP), 45.6% of patients with NICMP and 11.4% of patients with a previous ICD explantation due to device infection, the cohort of patients with an ICMP experienced the highest rate of VA: 9.6% of VA in the allover population compared with 16.7% in patients with ICMP, 3.8% of patients with NICMP, and 15.4% of patients with a previous device infection. The authors suggested a high risk of VA in patients with ICMP, especially of patients with a recent acute myocardial infarction [27].

Following these data, the randomized-controlled “VEST” trial (Vest Prevention of Early Sudden Death Trial) aimed to prove the benefit of WCD prescription on arrhythmic mortality in patients with severe left ventricular dysfunction (LVEF ≤ 35%) after acute myocardial infarction, with the result still being discussed very controversially: the data concerning the primary outcome of arrhythmic mortality could not show a benefit of WCD prescription (1.6% in the device group vs. 2.4% in the control group—*p* = 0.18), while the secondary outcomes including non-arrhythmic mortality and death from any cause differed statistically significantly between the groups (3.1% in the device group versus 4.9% in the control group, *p* = 0.04) in the as-treated analysis. The authors stated that the results for non-arrhythmic and total mortality were not corrected for multiple testing and were interpreted as a chance finding [30]. The discussion around low compliance and cross-over within the cohorts (WCD and optimal medical therapy versus optimal medical therapy alone) led to a post hoc analysis, with the authors stating a benefit on arrhythmic mortality in patients treated per protocol [31]. The data are further discussed in Section 3.1.

Based on the current evidence at that timepoint, the 2015 ESC Guidelines for the Management of Patients with Ventricular Arrhythmias and the Prevention of Sudden Cardiac Death (2) recommends the evaluation of a WCD in adult patients with poor left ventricular (LV) systolic function with a temporary risk of SCD who are not candidates for an ICD. The guideline sets a Class IIb Level of Evidence (LoE) C recommendation for patients that need to be bridged to heart transplantation or to a transvenous ICD implantation or in peripartum cardiomyopathy (PPCMP) and a Class IIa LoE C recommendation in patients recovering from an inflammatory heart disease with residual reduced ejection fraction.

The WCD use after acute myocardial infarction should be considered in selected patients with a high SCD risk due to incomplete revascularization, having a pre-existing LV dysfunction or arrhythmias more than 48 h after the index myocardial infarction (Class IIb recommendation, LoE C). However, these guidelines do not incorporate data from the VEST trial [30], which were published 3 years after the guidelines were introduced [2].

The WCD not only can bridge the time to a definite ICD candidacy as recommended by the guidelines but also may protect patients that suffer from ICD electrode infection or material failure with the need for a lead extraction, avoiding reimplantation during the same procedure, in order to minimize the risk of a new infection [32]. During the period between explantation and re-implantation, the patients may be spared from long hospital stays, which may have a positive impact on the healthcare system from an economical point of view. Although there are no scientific data on this topic, it might have a positive impact on concerned patients by not being hospitalized for several weeks, so they can keep up their usual levels of physical activity, can stay active in their working life, and can stay within their usual social environment.

There are no precise contraindications for WCD use specified in the current guidelines. According to the ESC Guidelines for the Management of Patients with Ventricular Arrhythmias and the Prevention of Sudden Cardiac Death [2] as well as the AHA/ACC/HRS Guideline for Management of Patients With Ventricular Arrhythmias and the Prevention of Sudden Cardiac Death (10), a survival of more than one year with a reasonable quality of life and a good functional status is assumed for an ICD candidacy; vice versa, if these prerequisites are not fulfilled an ICD would not be indicated. As there are no equivalent recommendations for the use of a WCD, national societies have published advisories to overcome the lack of specific contraindications in the guidelines: the American Heart Association (AHA) advices not to use a WCD in patients with a high risk of a nonarrhythmic fatal event that significantly exceeds the risk of an arrhythmic death, with an expected survival of less than 6 months (Class III recommendation, LoE C) [33]. The German Society of Cardiology specifies contraindications in a position paper for WCD use [34]: patients refusing an ICD implantation as well as patients not capable of handling a WCD or permanently not being eligible for an ICD candidacy according to the current guidelines (2,10) may likewise not be eligible for WCD prescription.

## 3. Multiparameter Monitoring

### 3.1. Monitoring Compliance

A very important factor for the effectiveness in preventing SCD and an important factor in the individual decision-making of prescribing a WCD is the daily wearing compliance. The wearing duration is counted if there is at least one of two ECG leads capturing electrical activation. The exact wearing duration is constantly monitored and captured as hours per day and can be accessed by the prescribing physician via the online platform “ZOLL Patient Management Network” provided by the manufacturer (ZOLL, Pittsburgh, PA, USA); see Figure 5 and Figure 6.

The *y*-axis shows the 24 h of each prescription day starting with 12:00 a.m. until 12:00 p.m., while every single day of prescription is plotted on the *x*-axis. The blue columns depict the time that the patient wears his/her WCD, and the interruptions in the columns or days without a column as seen in Figure 6 show the time when the WCD is not worn. The color of the columns does not have a specific meaning but alternate in color when a new month starts.

The position paper for WCD use from the German Society of Cardiology recommends not only intense patient education but also active surveillance of the actual wearing compliance. If the compliance is constantly lower than 20 h/day and does not improve even after a follow-up training, the ongoing use of a WCD is contraindicated [34].

Real-world data show a good compliance with a median wearing duration of 21.3–23.5 h in all patients and >20 h/day in most patients in big nationwide registries [17,20,35,36]. Factors such as a younger age of the patient have been identified to decrease wearing time [18,36], while neither gender, BMI, nor the number of inappropriate alarms showed an impact on the compliance [36].

In the clinical situation of patients after acute myocardial infarction with left ventricular dysfunction, the VEST trial [30] showed negative results in its’ primary outcome arrhythmic death; see Section 2.2. The results showed a below-average wearing compliance: in the VEST trial device cohort, only 53% had an average wearing duration ≥22 h within the planned 90 days of prescription, and 30% of patients stopped wearing the WCD within one month of randomization; 43% stopped within two months; and altogether, 80% stopped wearing the WCD earlier than the intended 90 days follow-up period, with 34% of patients not having worn the WCD at any time, resulting in 9/25 patients who did not wear their WCD being deceased during the follow-up as ventricular arrhythmias could not be detected or treated [13,19].

In the per-protocol analysis of the VEST data, the authors suggested that the SCD risk is decreased by the WCD in patients with a good wearing compliance (>90%, 21.6 h) compared with the whole cohort, which showed a worse compliance as explained above [31].

Technological advancements (reduced size and weight of the WCD) as well as thorough education might enhance the patient’s compliance, which is essential for benefiting from WCD prescription.

### 3.2. Monitoring Arrhythmias

#### 3.2.1. Sensor Interferences

Some patients may already be implanted with a cardiac pacemaker for bradyarrhythmia before WCD prescription. As the QRS complex may be deformed and imitate left bundle branch block, this can lead to misclassification of supraventricular arrhythmias. Additionally, paced QRS complexes can mislead the detection algorithm of the WCD into identifying SVTs as VTs, finally leading to inappropriate arrhythmia detection, triggering a significant number of alarms and inappropriate shocks in rare cases when patients are not able to abort the treatment [36]. In the case shown in Figure 7, a patient experienced total AV-block after receiving a WCD shock for ventricular tachycardia and deactivated inappropriate detections by pressing the response buttons depicted by the small pictures below the ECG stripe.

The pacing spikes of unipolar pacing can result in oversensing and can result in detected heart rates that do not correspond to an effective heart rate. When the ventricle is paced, the t-wave voltage increases in the same way as the QRS complex and can enhance oversensing as well. With a unipolar pacing program, up to 10% suffer from misled VA detection and are in danger of an inappropriate shock when not manually withholding imminent WCD therapy [37]. T-wave oversensing may also occur in patients with an intrinsic rhythm such as that shown in a patient in Figure 8.

Clinical experience in this cohort of patients shows that bipolar ventricular pacing may also lead to inappropriate WCD alarms [38].

The most important reason for interference in inadequate automatically triggered alarms is artefacts for 95.6% in the Austrian WCD registry [39], mostly coming from patient movements resulting in artefacts or having poor skin-to-electrode contact while having an underlying sinus rhythm, such as that shown in Figure 9.

In an analysis of 106 patients (median 52 years (P25: 37 years; P75: 66 years), 31% female), neither compliance, gender, having a previous cardiac implantable electronical device (CIED) and/or experiencing active pacing, being diagnosed with other arrhythmias, QRS duration, BMI, nor age were predictors of inappropriate alarms [36]. Data from a German single center experience reported by Erath et al. observed that 57% of patients have inappropriate alarms due to artefacts. The data suggested that skinny and more active patients trigger a significantly higher number of alarms resulting from insufficient skin contact of the ECG electrodes. Within the publication, the proposed solution was to reprogram the VT zone from the preset 150 bpm to 180 bpm based on the MADIT-RIT trial, which originally investigated reprogramming CIED to avoid inappropriate therapies [40,41].

When patients are properly educated and are able to react adequately, inappropriate shocks can usually be avoided by pressing the response buttons, resulting in a low inappropriate shock rate, ranging from 0.4% (26/6043) in the German WCD registry [18] to 0.5% (10/927) in the prospective WEARIT-II registry (17) and 0.8% (7/879) of patients in our Austrian WCD registry [39], with up to 1.9% (2/106) of patients having inappropriate treatments in real-world cohorts with a smaller number of patients [36].

#### 3.2.2. The WCD as a Shock Box

The WCD provides a therapeutic option to prevent SCD: after adequate detection of VA, patients have the option to react to an acoustic and vibrational alarm to either reject the WCD shock by pressing the buttons or, if not reacting, to receive a shock from the WCD. Appropriate shocks are meant to be delivered as a reaction for VAs (see Figure 10), and shocks for supraventricular arrhythmias, atrial fibrillation, atrial flutter, or normal sinus rhythm with artefacts are considered inappropriate. That a shock is not always necessary even though a VA is present was shown in the WEARIT-II registry: 22 patients aborted the treatment in 90/120 (75%) sustained VT events opposed to 30/120 (25%) of events requiring therapy due to hemodynamic instability (17).

As registry data and data from clinical trials show, the efficacy of WCD shocks is usually very high (94 to 100%) (15, 17–19), but there might still be the need for mechanical cardiopulmonary resuscitation in patients with VA: if a patient is in a ventricular storm, the electrical therapies applied by the WCD might be not enough; additionally, the WCD can also detect non-shockable rhythms such as asystole that also result in cardiac arrest. The events are captured and sent to the online network, and consequences such as optimization of medical therapy, planning of a catheter ablation, etc. can be drawn from the transmitted information to a later timepoint.

Besides that, there is a German initiative to enhance the early transmission of events from patients with potential life-threatening arrhythmias to optimize the rescue chain to increase the likelihood of a survival after cardiac arrest. This initiative is working on establishing a telemedical link from the WCD directly to local emergency call centers and/or to local healthcare practitioners so that advanced life support can be provided if the WCD shock is not successful or the ECG shows a significant recording that does not require defibrillation such as bradycardia or asystole but may be a fatal event as well and cannot be cured with an electrical shock [42].

#### 3.2.3. Manually Triggered Alarms

In addition to automatically triggered alarms as described above, the patients can trigger an ECG recording themselves when they feel palpitations or any symptoms that may be associated with heart rhythm disorders.

In the Austrian WCD registry, 5492 manually recorded ECGs were triggered by 555/879 (63%) patients. Within these ECGs, only one ECG showed slow sustained V and 25 ECGs in nine patients showed non-sustained VTs. Twenty-six patients experienced atrial fibrillation in 81 events, four patients recorded an ECG due to 5 bradycardic events, 2 ECGs in two patients showed premature ventricular beats (PVC), and 42 ECGs in nineteen patients showed supraventricular tachycardia (SVT). In 11 patients (42 ECGs), sinus tachycardia was detected, and in 96.6% (5308/5492), the triggered ECGs showed normocardic sinus rhythm [39].

Manually recorded ECGs can help detect non-VA arrhythmias such as new onset atrial fibrillation or relevant bradycardia that is not detected by the WCD algorithm but needs an experienced cardiologist to interpret ECG recordings in the clinical context.

### 3.3. Monitoring Heart Failure

Besides patients with inherited cardiomyopathies or channelopathies who often present with arrhythmias without clinical evidence of systolic dysfunction, a high percentage of patients prescribed with a WCD suffer from heart failure with reduced ejection fraction (HFrEF). Besides arrhythmia monitoring, monitoring heart failure and signs for an imminent decompensation could help reduce morbidity and mortality in this dominant cohort of patients at risk for SCD. The risk of re-hospitalization after an index event of an admission for symptomatic heart failure is greatest within the first 30 days after discharge [43,44].

Preceding an imminent cardiac decompensation that requires an inpatient admission to a hospital, patients themselves can notice early signs of an already manifest cardiac decompensation [45]. Symptoms such as an increased burden of atrial arrhythmias, an increase in resting heart rate or decreased physical activity as well as increased lung fluid levels can be detected by ICD algorithms. Remote monitoring with implantable devices to anticipate a manifest decompensation before clinical symptoms are present has shown improved clinical outcomes in studies with several hundred patients. In HFrEF, a monitoring approach that uses several parameters captured with an ICD is mentioned with a Class IIb recommendation, LoE B in symptomatic patients in the current guidelines for heart failure [46] In HFrEF and HF with preserved ejection fraction (HFpEF), measurements of pulmonary arterial pressures with the CardioMems system may be considered (Class IIb recommendation, LoE B) in the specific situation with a previous HF hospitalization to avoid re-hospitalization due to HF [47,48,49,50].

These implantable devices are mentioned in the current ESC heart failure guidelines [50] and can measure surrogate parameters or direct invasive pressures depending on the type of device and may help to monitor the course of the disease and to intervene by anticipating cardiac decompensations before hospitalization is needed.

The population provided with a WCD is at risk of acute heart failure as patients are newly diagnosed with a CMP or are in an acute phase of a cardiac disease, and very often, the prescription of a WCD occurs in an inpatient setting. The idea to monitor so-called cardiac acoustic biomarkers (CAB) via sensors on the harness of the WCD came up to identify early evidence of cardiac decompensation in patients with HFrEF and specifically in patients with a left ventricular ejection fraction (LVEF) ≤ 35% in the heart sounds registry, the “HEAR-IT” study [51] A multiparameter monitoring algorithm was applied on 671 patients (61 ± 13 years) prescribed with a WCD: the cardiohaemic vibrations measured with the defibrillation electrodes adjacent to the patient’s body surface incorporating a three-axis accelerometer and the simultaneously registered two lead surface ECGs. From these sensors, the heart sounds are combined with the information for the ECG, and a combination of the electromechanical activation time (EMAT) and the third heart sound (S3) strength was measured over time; the trend of this combined parameter showed a good correlation to classify patients into groups being at either low or high risk for a heart failure event.

The EMAT is a parameter measured from the onset of the Q wave on the surface ECG to the peak of the first heart sound and serves as parameter for systolic function if prolonged. The third heart sound is measured on a scale of 0–10 and is also a well-known sign for heart failure in clinical auscultation and is a sign for increased intracardiac filling pressures. The algorithm used a 10 s measurement every 5 min to measure the CABs and the heart rate. These values were observed over time, and thresholds were defined. Patients with CABs above the upper threshold were at very high risk for a heart failure event while, if the CABs improved, they needed to fall below the lower cut-off to count to the low-risk cohort. During follow-up, 81/671 (12%) had a heart failure event (cardiovascular death, arrhythmias and hospital admissions, and emergency department visits for HF). The increase in CABs above the upper threshold identified 69% of the events at least 2 weeks before and 90% of HF events at least 3 days before. The initial classification within the first 7 days after discharge from the index hospitalization in high- or low-risk HF events through CABs was more accurate in the prediction of an event compared with the NYHA classification, while on the other hand, the algorithm combining CABs and HR had a negative predictive value of 94% for HF events. The authors proposed the integration of CABs in clinical practice to help prevent HF events [51] Currently, CABs recorded by the WCD are not available for real-world patient care and were only studied in trials.

An additional feature of the newer WCD generation, called the “TRENDS” option provides a combined multiparameter monitoring approach for heart failure incorporating the heart rate and changes in average heart rate, the activity via a daily step count as well as the body orientation/position detected via a three-axis accelerometer. All of this information is available as daily data and was captured over time as trends in the online WCD network; see Figure 11.

The TRENDS data were captured and available also in clinical patients outside of clinical trials, while the visibility within the online network needs to be activated for each patient and each physician separately and is not routinely used for follow-ups. Recently, a retrospective analysis of TRENDS data patients from the PROLONG II trial [52] proposed a clinical application of the TRENDS data: 267 patients (31.9% female, mean LVEF 25.3 ± 8.5%) with a newly diagnosed CMP were observed over the time of WCD prescription (111.8 ± 74.5 days). The first and last seven days of usage were compared and showed significant changes in heart rate, step count, and five-minute heart rate variability approximate (HRV5), which is a surrogate for beat-to-beat heart rate variability and was calculated from the data available from the online network. After multivariate analysis, the change in HRV5 seemed to be an independent predictor for LVEF improvement defined as an increase of ≥ 10%. The authors stated that HRV5 may be a potential indicator of treatment response during the evaluation phase of newly diagnosed heart failure [53].

The physical activity (PA) has a prognostic role for HF events and cardiovascular death in patients at a high risk as proven in a cohort of post-myocardial infarction patients with patients self-reporting their physical activity [54] and in studies with device-measured physical activity in heart failure patients [55,56].

A retrospective, observational study analyzed the average daily step count and a change within 4057 patients. These surrogate parameters of physical activity were measured by the WCD, and whether a decrease in PA is prognostic for a deterioration of heart failure and ventricular arrhythmia events was evaluated. Patients with a lower step count, with a cut-off of 3637 steps per day during the first week after WCD prescription, showed a 4.3 times higher likelihood to receive a shock for VA compared with more active patients, especially in the first month after WCD prescription. Similar to the CABs [51], a low PA within the first seven days after hospital discharge from the index hospitalization was associated with a higher event rate [13]. In a study evaluating 4928 female patients, a decline in PA was seen 2 weeks before a WCD treatment for VA [57] In contrast, all studies about PA in patients after an index HF event attested to the feasibility of telemedical surveilled step count in patients with an overall good compliance as a possible tool for remote monitoring in heart failure [13,57,58].

While the daily step count is a surrogate for physical activity, the 6 min walk test (6MWT) is a well-established and objectifiable test for the functional status of a heart failure patient. In times with a rise in importance of telemedical options, reliable and objectifiable options are necessary. A randomized clinical trial confirmed the feasibility of a 6MWT, analyzing WCD-guided 6MWT tests performed regularly at home in 197 patients (57 ± 12 years, LVEF 23 ± 7%) over a run of 8 weeks. There was no difference within these groups.

Currently, these data are not routinely used although they are partially available for the treating physicians, as the clinical evidence not only to detect but also to intervene and prevent HF events through any of these noninvasive diagnostic options is still missing.

### 3.4. Monitor Therapy Success

In the acute phase of a newly diagnosed CMP, the establishment and up-titration of pharmacological therapy is essential [50] Heart rate monitoring as permanently measured by the WCD can be useful in patients with atrial fibrillation to assess a pharmacologically established rhythm or rate control or to assess target values in heart rate. Case reports already reported clinical application of the WCD as a monitoring device to supervise up-titration of HF medication and to control the heart rhythm as well as to consequently take action on repeated onset atrial fibrillation to avoid cardiac decompensation [59].

An upcoming study will investigate betablocker up-titration in heart failure patients monitored with a WCD. Beta blockers may have bradycardia as a side effect and do not unfold their full therapeutic effect when under-dosed. Consequently, the WCD is used as a monitoring tool during up-titration of beta blocker therapy. (“Optimizing Beta Blocker Dosage in Women While Using the Wearable Cardioverter Defibrillator—OPT BB women”, clinicaltrials.gov identifier: NCT04504188).

Following the current ESC heart failure guidelines [50], the decision about implanting an ICD should be made after implementation of an optimal medical therapy (OMT) for three months while patients are at risk for SCD and seem to be unprotected. Monitoring therapy success and protecting patients from SCD in order to avoid untimely and unnecessary ICD implantations during up establishment and up-titration of heart failure therapy was tested in the PROLONG study [60], with 156 patients (54 ± 14 years, 35% female) with a LVEF 24 ± 7% being prescribed with a WCD after being newly diagnosed with HFrEF. A follow-up was scheduled after 3 months to assess the LVEF and functional status of the patients. Patient in the PROLONG study received a WCD for NICMP in 55%, 29% had an ICMP, 12% had a PPCMP, and 4% of patients were diagnosed with acute myocarditis. After 3 months, the LVEF and functional status of the patients were reassessed and a decision on whether to prolong the evaluation was made. If the LVEF was between 30 and 35% and the LVEF improved by ≥ 5% within the first three months or if the maximally tolerated guideline-directed medical therapy was not established optimally yet, the WCD prescription was prolonged for another 3 months and a reassessment was scheduled. While 65/156 (42%) showed a LVEF above 35% after 3 months, another 26/156 (19%) improved after 6 months of therapy in total.

A total of 11/156 (7%) patients experienced 12 WCD shocks after a median prescription duration of 59 days (13;161 days), with ten shocks happening within the first three months after WCD prescription and two shocks being observed between the third and the sixth months. The authors therefore conclude that malignant ventricular arrhythmias may occur throughout the course of the 6 months and may enhance the high risk of SCD in this population [60].

## 4. Conclusions

The WCD is an established treatment option in selected patient groups with a high risk of SCD who are not eligible for an immediate ICD implantation. The prescription of a WCD may help to avoid unnecessary ICD implantation and can cover the period of establishing an optimal medical treatment.

Besides the potential benefit for patients and healthcare systems, the patient is the key component to ensure therapy success with a WCD: the downside of a wearable device compared with an implanted device is that the WCD can only protect from malignant ventricular arrhythmias when actively worn. Through the online network provided by the manufacturer, the treating physician can actively surveil the wearing duration day by day and, similar to ICD telemedical approaches, receive alarms when compliance is insufficient.

Another potential downside from the vest device is the arrhythmia detection and treatment algorithm, which produce a high number of inadequate alarms. The false alarms originate primarily from motion artefacts and usually can be handled by the patients after thorough education about their device. Sensor interferences such as pacing artefacts from implanted pacemakers do not pose a big issue in clinical routine after years of WCD use. As comparable in CIED programming the indication, the medical history and potential sensor interferences need to be taken into account when prescribing a WCD and when programming individual thresholds. Nevertheless, data from big nation-wide registries have shown a low number of inappropriate shocks following these inadequate alarms and the WCD has been proven to be safe and effective in detecting and treating malignant arrhythmias.

Additionally, functions to assess the functional status via surrogate parameters to detect a deterioration in heart failure can support healthcare providers in identifying subgroups with a high cardiovascular risk or even in differentiating between therapy responders and non-responders early and potentially giving the possibility to intervene early and to prevent imminent CV events, while the evidence for a clinical benefit from randomized controlled trials is still missing. Generally, a WCD produces a large amount of data through its multiparameter monitoring function, which needs to be thoroughly reviewed. Features such as the TRENDS data and CABs that provide information about functional status and physical activity could be added as part to the treatment and follow-up of heart failure patients, while for now, this information is not yet integrated in routine clinical practice.

Another big downside of more than two decades of WCD use is that data from randomized-controlled clinical trials are still scarce, and the evidence is based mostly on registry data [61].

As an outlook into the future, cardiovascular sensors such as the WCD not only can help in protecting from SCD but also in guiding medical therapy for heart failure patients. The current pandemic situation helps to accelerate the establishment of technical infrastructure and the financial reimbursement of telemedicine via trained medical staff, and these functions could potentially help to reduce ambulatory follow-ups and hospitalizations in the future to protect patients and to lower the economic healthcare burden.

## Figures and Tables

**Figure 1 sensors-22-00022-f001:**
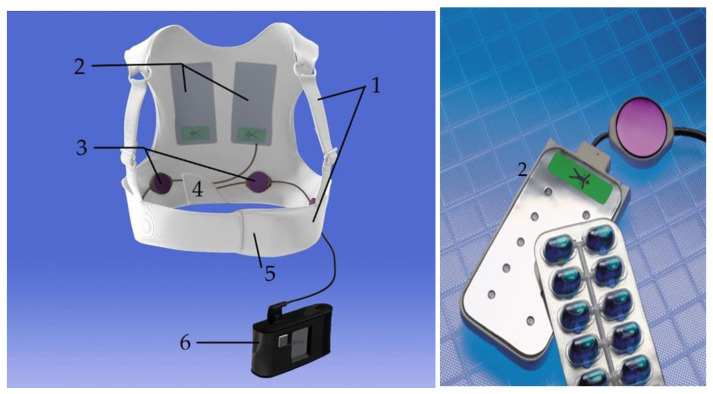
A WCD with its components: the fabric garment with an adjustable belt and shoulder straps (**1**), self-gelling defibrillation pad with ten gel capsules (**2**), the electrode belt (**3**), the vibration box (**4**), the heart sounds sensor included in the apical defibrillation pad (**5**), and the monitor with the response buttons (**6**); © ZOLL CMS GmbH.

**Figure 2 sensors-22-00022-f002:**
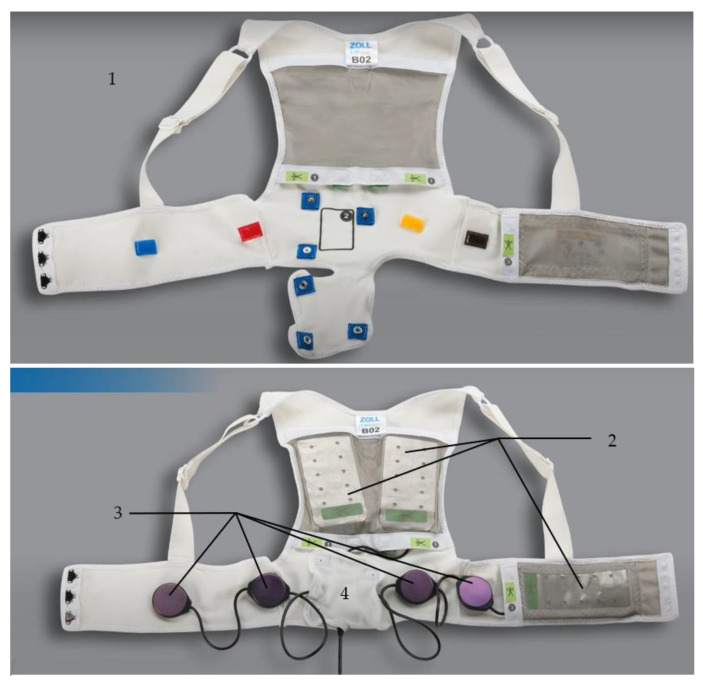
Upper picture: The fabric garment without the technical components (**1**). Lower picture: the fabric garment with the defibrillation pads (**2**), the electrode belt (**3**), and the vibration box (**4**); ©ZOLL CMS GmbH.

**Figure 3 sensors-22-00022-f003:**
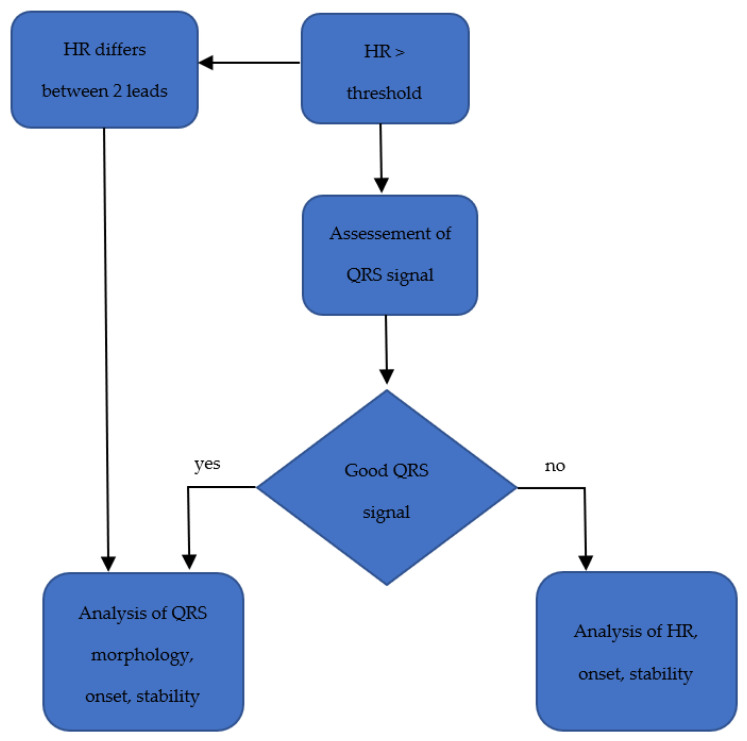
Arrhythmia detection algorithm.

**Figure 4 sensors-22-00022-f004:**
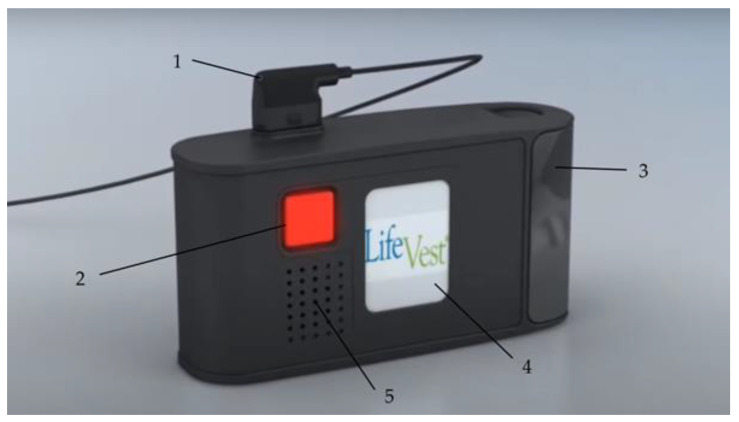
The monitor unit consists of the connection point to the electrode belt (**1**), the response buttons (**2**), the rechargeable battery (**3**), the LCD touch screen (**4**), and a speaker (**5**); ©ZOLL CMS GmbH.

**Figure 5 sensors-22-00022-f005:**
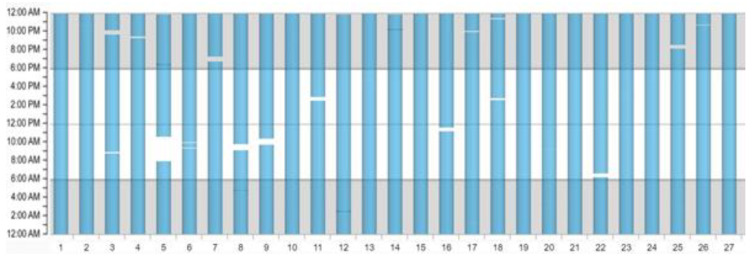
ZOLL Patient Management Network depicting a patient with >23 h/day.

**Figure 6 sensors-22-00022-f006:**
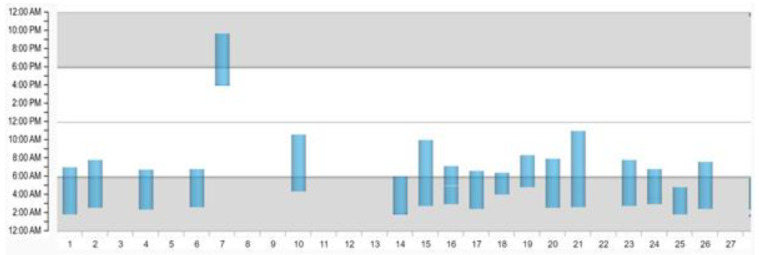
ZOLL Patient Management Network depicting a patient with 5.3 h/day wearing.

**Figure 7 sensors-22-00022-f007:**
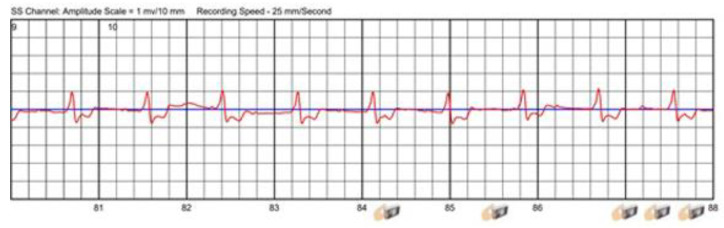
Pressing the response buttons while experiencing AVB III°.

**Figure 8 sensors-22-00022-f008:**
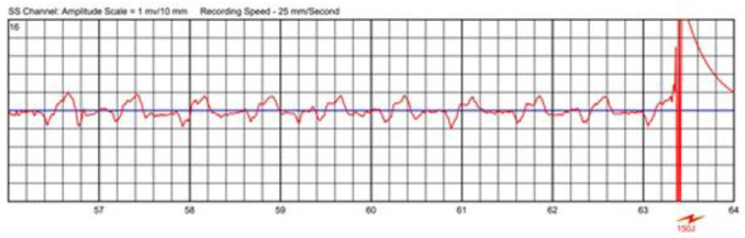
T-wave oversensing leading to an inappropriate shock.

**Figure 9 sensors-22-00022-f009:**
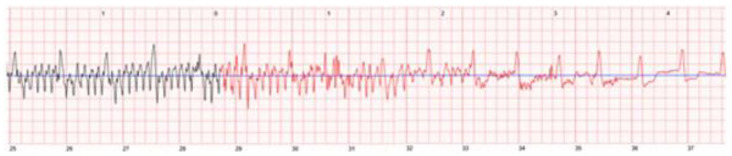
Automatically recorded ECG—artefacts with underlying sinus rhythm.

**Figure 10 sensors-22-00022-f010:**
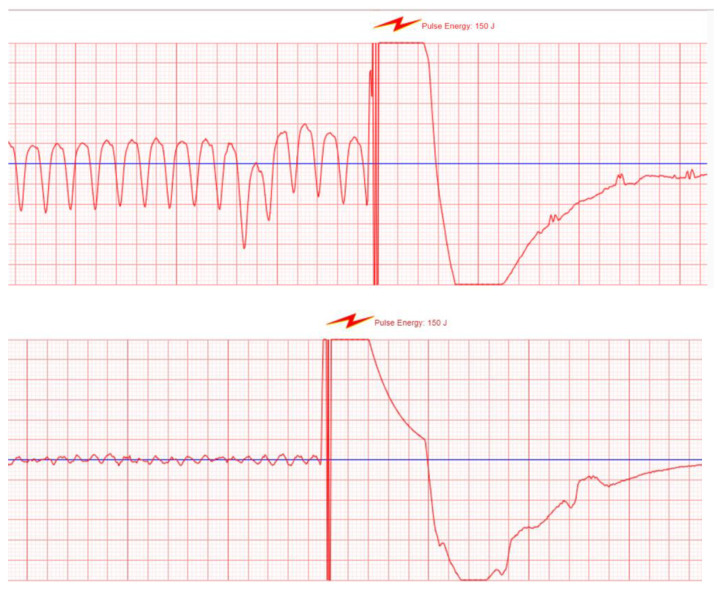
WCD shock for VF and VT.

**Figure 11 sensors-22-00022-f011:**
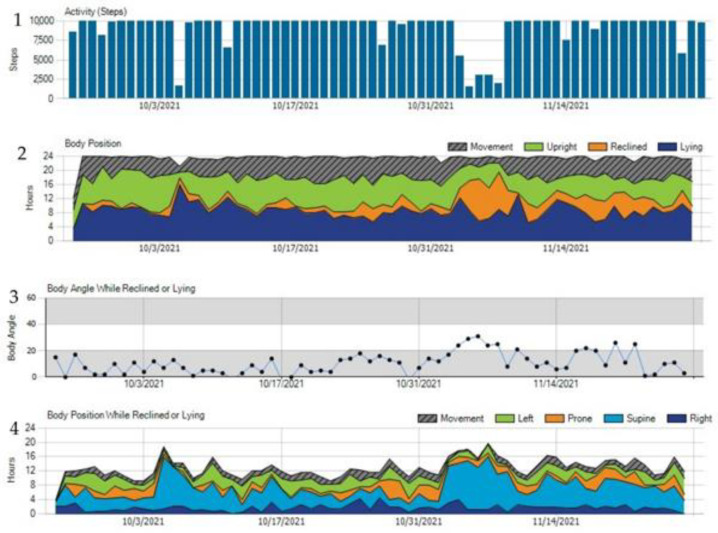
TRENDS data of a patient with an ongoing WCD prescription showing the daily step count (**1**), the body position (**2**), the body angle while reclined or lying (**3**) and the body position while reclined or lying (**4**).

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
