# Peer review of "Multiparameter Monitoring with a Wearable Cardioverter Defibrillator"

_sensors, 2021, doi:10.3390/s22010022_

Round 1
Reviewer 1 Report
This review manusrcipt is presenting an analysis of existing publications on Wearable Cardioverter-Defibrillators (WCD) that is present on the market with only one manufacturer (ZOLL Inc.).
It provides an additional insight compared to the latest review published, from Cheung, Christopher C., Jeffrey E. Olgin, et Byron K. Lee. 2021. « Wearable Cardioverter-Defibrillators: A Review of Evidence and Indications ». Trends in Cardiovascular Medicine 31 (3): 196‑201. https://doi.org/10.1016/j.tcm.2020.03.002.
(which publication could be cited)
Explanations related to passed or on-going works using additional biomarkers (acoustic, physical activity) are given.
It is not clear how much these biomarkers are integrated into the WCD device and how easy is the interpretation of these informations by the practitionneer.
Some points need to be addressed:
Several abreviations need to be defined: CMP, DGK, AMI, CIED
Line 205-207:
It is mentionned "the ischemic cohort" whereas the end of the sentence relates to ICMP. This might be rephrased.
Line 237-239:
Are we sure that patient with contraindications to an ICD are not eligible to WCD prescription. If this is the case, an additional sentence needs to clarify this fact.
From line 263:
This paragraph relates to the "VEST Trial", including some description of the trial. Nevertheless, this trial has already been mentionned before in the text.
It is necessary to shift some partsof the ptext in order to avoid confusion.
Line 431:
The text "is a sign increased intracardiac filling pressures" is probably missing one word.
Lines around "monitoring heart failure" :
Explanation about the level of integration of the said biomarkers should be disclosed for clarity.
Reviewer 2 Report
see the attached file

Round 2
Reviewer 2 Report
The authors are to be congratulated on working to make revisions to the text. I appreciate the authors efforts and the changes made that improved the overall quality of the manuscript, particularly concerning the message the authors provided. A few comments are listed below:
1) Line 368: “In an analysis of...,” this sentence seems uncomplete.
2) The case suggested (Neglected lead tip erosion: An unusual case of S-ICD inappropriate shock. J Cardiovasc Electrophysiol. 2020 Dec;31(12):3322-3325, doi: 10.1111/jce.14746) is a representative case of potential use of a wearable cardioverter defibrillator. In fact, in cases as the one mentioned, a wearable cardioverter defibrillator could be an interesting solution for delay the device reimplantation, further minimizing the risk of re-infection. Please,cite the case suggested to corroborate this message.
Line 256-257: “..may also protect patients that suffer from ICD electrode infections or material failure with the need of a lead extraction”. I suggest to rephrase this sentence as follow, citing the above-mentioned manuscript:
“…may also protect patients that suffer from ICD electrode infection or material failure with the need of a lead extraction, avoiding the reimplantation during the same procedure, in order to minimize the risk of a new infection”
Author Response
Please see the attachment.

This manuscript is a resubmission of an earlier submission. The following is a list of the peer review reports and author responses from that submission.